# Molecular Mechanisms of Inhibitory Effects of Bovine Lactoferrin on Invasion of Oral Squamous Cell Carcinoma

**DOI:** 10.3390/pharmaceutics15020562

**Published:** 2023-02-07

**Authors:** Chanbora Chea, Mutsumi Miyauchi, Toshihiro Inubushi, Kana Okamoto, Sivmeng Haing, Takashi Takata

**Affiliations:** 1Department of Oral & Maxillofacial Pathobiology, Graduate School of Biomedical & Health Sciences, Hiroshima University, 1-2-3 Kasumi, Minami-ku, Hiroshima 734-8553, Japan; 2Department of Orthodontics and Dentofacial Orthopedics, Graduate School of Dentistry, Osaka University, 1-8 Yamada-Oka, Suita 565-0871, Japan; 3Shunan University, 843-4-2 Gakuenndai Syunan, Yamaguchi 745-8566, Japan

**Keywords:** lactoferrin, oral squamous cell carcinoma, AP-1, MMP-1, MMP-3, invasion

## Abstract

Lactoferrin (LF), an iron-binding glycoprotein, has been reported to have anticancer properties. However, the molecular mechanisms behind its anticancer effects on oral squamous cell carcinoma (OSCC) have not yet been elucidated. Therefore, we aimed to clarify the effects of LF on invasion of OSCC, and its underlying molecular mechanism. OSCC cell lines, HSC2 and HOC313, were treated with bovine LF (bLF). The effects of bLF on cell invasion were examined by a chamber migration assay, wound healing assay, and Boyden chamber method with a basement-membrane-analogue. Expression levels of MMP-1, MMP-3, and AP-1 were examined using RT-PCR, qRT-PCR, and western blotting. Roles of LRP1, a receptor of bLF, on cell invasion were analyzed using siLRP1 knockdown cells. Furthermore, to clarify the importance of LRP1 in invasion, the effects of bLF on tPA-induced invasion of OSCC cells were examined. The invasion assays showed that bLF suppressed invasion of the OSCC cells. Moreover, bLF down-regulated AP-1, and resulted in reductions of MMP-1 and MMP-3. With SiLRP1 knockdown, OSCC cells failed to induce their invasion, and bLF was not able to exert its effects on invasion. Furthermore, bLF remarkably inhibited tPA-induced cell invasion. These findings suggest the importance of LRP1 in bLF-suppressed invasion of OSCC cells via the reduction of AP-1 and MMP production.

## 1. Introduction

Oral squamous cell carcinoma (OSCC) is one of the most common cancers worldwide and the survival rate of patients with OSCC has not significantly improved despite development and innovations in diagnosis and therapy [1]. Similar to most malignancies, OSCC cells show extensive invasion, which is the main characteristic of malignant tumors and the main cause of cancer related death [2]. One of the mechanisms involved in the invasion of OSCC cells is the destruction of basement membranes by overexpression of matrix metalloproteinases (MMPs) in cancer cells.

Two separate serine proteinases, urokinase and tissue plasminogen activator (uPA and tPA), are responsible for the activation of plasmin from the inactive proenzyme plasminogen. The invasion of cancer cells requires proteolytic degradation of the extracellular matrix by certain molecules including plasmin [3].

Lactoferrin (LF), an 80-kDa member of the transferrin family of iron-binding glycoproteins, is found in external secretions, predominantly in milk [4]. Recently, it has been reported to have various biological anti-inflammatory, anti-viral, anti-bacterial, and anti-tumor properties [5,6,7,8,9,10,11]. Interestingly, LF and its derivatives have been used in pre-clinical animal investigations, human clinical trials, and cancer therapy for many years [12,13,14,15]. Inhibition of invasion in cancer cells is one of the main mechanisms by which LF may inhibit cancer malignancies. Recent studies have shown that LF directly inhibits invasion in colorectal and breast cancers [16,17]. However, the effects of LF on invasion of OSCC, and its underlying mechanisms, have not been clearly elucidated. Hence, we examined the effects of LF on the invasive properties of OSCC cells, and its mechanisms at a molecular level.

## 2. Materials and Methods

### 2.1. Reagents

Bovine lactoferrin (bLF) was provided by Sunstar Inc., (Osaka, Japan). Distilled water was used to dilute bLF to a concentration of 10 mg/mL by vortexing and filtering through a 0.2 μm pore size filter. The solution was then stored at −20 °C. Non-enzymatic human tPA was purchased from Molecular Innovations Inc. (Novi, MI, USA).

### 2.2. Cells and Cell Culture

HNSCC cell lines were used in this study. HSC2 was obtained from the Japanese Collection of Research Bioresources Cell Bank and maintained in RPMI-1640 (Nacalai Tesque, Inc., Tokyo, Japan); the HOC313 cell line was supplied by Prof. Kamata (Hiroshima University, Hiroshima, Japan) and cultured with DMEM (Nissui Pharmaceutical Co., Tokyo, Japan). All medium was supplemented with 10% heat-inactivated FBS (biowest, Nuaillé, France) and 100 U/mL penicillin-streptomycin (Sigma Aldrich, St. Louis, MI, USA). Both cell lines were incubated at 37 °C in 5% CO_2_ in air.

### 2.3. Wound Healing Assay

Six hundred thousand HSC2 and HOC313 cells were grown in 6 cm tissue culture plates. After 24 h of incubation, the medium was changed and the cells were pre-treated with or without bLF (1, 10, or 100 μg/mL) for 72 h. Cells were then scratched and continuously treated with or without bLF. For both 20 nM tPA and 100 μg/mL bLF stimulations, the medium was incubated for 2 h with tPA and bLF prior to application into the culture dishes. A pen was used to mark the tissue culture plates for repeated images. The wound closure distances were calculated and compared with scratches at 0 h, which were considered as 100%.

### 2.4. Chamber Migration Assay

After trypsinization, 2.5 × 10^4^ cells of 72 h pre-treated HSC2 and HOC313 cells with or without bLF (1, 10, or 100 μg/mL), were suspended in 200 µL of medium and placed into the upper chamber of culture inserts of 8 μm pore size (Becton Dickinson Labware, Franklin Lakes, NJ, USA). The inserts were then placed in a 24-well cell culture plate with bLF (0, 1, 10, or 100 μg/mL). The cultures of HSC2 cells were incubated for 6 h, and HOC313 for 12 h, at 37 °C. After the incubation, cells that had migrated to the lower side of the membrane were fixed with 10% paraformaldehyde and stained with Mayer’s hematoxylin. The cells on the upper surface of the filter were wiped off with a cotton swab. The migration cells that attached to the bottom of the membrane were counted under a microscope at 200× magnification.

### 2.5. In Vitro Invasion Assay

HSC2 and HOC313 cells (6 × 10^5^) were cultured in 6 cm dishes and incubated for 24 h. The medium was changed prior to pre-treatment of the cells with or without bLF for 72 h. Culture inserts of 8 μm pore size (Becton Dickinson Labware, NJ, USA) were inserted into 24-well plates, which contained 0.5 mL of medium. The inserts were then coated with 100 μL of Matrigel (1 mg/mL; BD Bioscience, Bedford, MA, USA). HOC313 cells (2 × 10^4^) pre-treated with 100 μL of medium, in the presence or absence of bLF (1, 10, or 100 μg/mL), were added into the upper chambers and incubated at 37 °C. Stimulation with tPA (20 nM) and bLF (100 μg/mL) was performed in the same manner as for the wound-healing assay. The invasive cells were then fixed with 10% paraformaldehyde and stained with Mayer’s hematoxylin. The cells on the upper surface of the chambers were removed using a cotton swab. Finally, the membranes were separated from the insert and mounted on glass slides. The number of invading cells was counted under a microscope (200× magnification).

### 2.6. Reverse Transcription Polymerase Chain Reaction (RT-PCR)

The cells were collected with cold PBS. Total RNA was extracted from fresh cells using Tri Reagent (Molecular Research, Inc., Albany, NY, USA), purified using the RNeasy kit (Qiagen, Hilden, Germany) according to the manufacturer’s instructions, and quantified by using standard spectrophotometric methods. Briefly, complementary DNA (cDNA) was synthetized from 1 μg of total RNA using the ReverTra Ace kit (Toyobo Biochemicals, Tokyo, Japan). Primer sequences included the following: human LRP1, 5′-AGCAAACGAGGCCTAAGTCA-3′ (forward), 5′-GCTGCTTGTGCTGATGGTAA-3′ (reverse); human MMP-1, 5′-ATGCTGAAACCCTGAAGGTG-3′ (forward), 5′-CTGCTTGACCCTCAGAGACC-3′ (reverse); human MMP-3, 5′-GCAGTTTGCTCAGCCTATCC-3′ (forward), 5′-GAGTGTCGGAGTCCAGCTTC-3′ (reverse); GAPDH 5′-GCATCCTGGGCTACACTGAG-3′ (forward), 5′-TCCACCACCCTGTTGCTGTA-3′ (reverse).

Total cDNA was amplified with 1.25 U of rTaq-DNA polymerase (Qiagen, Hilden, Germany) using a thermal cycler (Astec, Fukuoka, Japan) for 30 cycles after an initial 30 s denaturation at 94 °C, annealing for 30 s at 60 °C, and extension for 1 min at 72 °C for all primer sets. The amplification products were electrophoresed on 1.5% agarose/TAE gels and visualized by ethidium-bromide staining under UV.

### 2.7. Quantitative RT-PCR (qRT-PCR)

The expression of MMP1 and MMP3 was quantified by qRT-PCR analysis. Total RNA and cDNA were prepared as above. Briefly, cDNA was amplified with a Kapa SYBR FAST qPCR kit (Kapa Biosystems, Inc., Wilmington, MA, USA) with primers for human MMP-1 5′-GGCCCACAAACCCCAAAAG-3′ (forward), 5′-ATCTCTGTCGGCAAATTCGTAAGC-3′ (reverse); human MMP-3 5′-GATGCCCACTTTGATGATGATGAA-5′ (forward), 5′-AGTGTTGGCTGAGTGAAAGAGACC-3′ (reverse); GAPDH 5′-GGCCTCCAAG-GAGTAAGACC-3′ (forward), 5′-AGGGGTCTACATGGCAACTG-3′ (reverse). The results were elucidated by a StepOnePlus Real-Time PCR system, StepOne Software, version 2.1 (Applied Biosystems, Tokyo, Japan).

### 2.8. Western Blotting Analysis

The cells were harvested by washing with ice-cold PBS and lysed in buffer containing 0.1% Triton X-100 (Roche, Castle Hill, Australia), 10 μg/mL L-1 chlor-3- (4-tosylamido)-4 Phenyl-2 butanone (TPCK), 1 mM DTT, 0.1 mM Na3VO4, 10 μg/mL L-1 chlor-3-(4-tosylamido)-7-amino-heptanon-hydrochloride (TLCK), 0.1 mM leupeptin, and 50 μg/mL phenylmethylsulfonyl fluoride (PMSF) for 30 min. Lysates were centrifuged at 16,000× *g* for 20 min at 4 °C. The amount of protein in each sample was measured by using the Bradford protein assay (Bio-Rad, Richmond, CA, USA), and samples were then heated in 4× Laemmli buffer at 100 °C for 3 min, fractionated by 10% polyacrylamide gel electrophoresis (PAGE), and transferred onto nitrocellulose membranes (Schleicher & Schuell, Dasse, Germany). The membranes were then blocked using 3% milk for 1 h at room temperature, inoculated with primary antibodies, and kept overnight at 4 °C. The following antibodies were applied: p-ERK1/2 (4376; diluted to 1:1000), ERK 1/2 (4695; diluted to 1:1000), p-c-Fos (5348; diluted to 1:1000), c-Fos (2250; diluted to 1:1000), p-JNK (4668; diluted to 1:1000), JNK (9258; diluted to 1:1000), p-c-Jun (3270; diluted to 1:1000), c-Jun (9162; diluted to 1:1000), LRP1 (2703-1; diluted to 1:7000), MMP-1 (R&D system, MN5541; diluted to 1:1000), MMP-3 (sc-21732; diluted to 1:1000), and β-actin (A2228; diluted to 1:8000). The membranes were then incubated with secondary antibodies (diluted to 1:1000) for 1 h at room temperature. The ECL western blotting detection system (GE Healthcare, Buckinghamshire, UK) (Amersham, Piscataway, NJ, USA) was used for visualization.

### 2.9. Silencing by Small Interfering RNA (SiRNA)

The human lipoprotein receptor-related protein-1 (hLRP1-1) with sense siRNA, GCAGUUUGCCUGCAGAGAUtt, antisense siRNA, AUCUCUGCAGGCAAACUGCtt, and hLRP1-2 with sense SiRNA AUGCUGACCCCGCCGUUGCtt, and antisense siRNA GCAACGGCGGGGUCAGCAUtt, were purchased from Applied Biosystems. Four microliters of siRNA, gene-specific siRNA oligomers (20 nM), and 5 μL of RNAiMAX (Invitrogen, Waltham, MA, USA) were respectively diluted into 250 μL of Opti-MEM (Invitrogen). After a 20 min incubation at room temperature, the complexes were added into 2 × 10^5^ of HSC2 and HOC313 cells and plated into 3.5 cm dishes. The dishes were adjusted to a final volume of 2 mL of medium. After 24 h of incubation at 37 °C in 5% CO_2_ in air, the medium was changed. The cells were stimulated with or without tPA or bLF.

### 2.10. Statistical Analysis

All numerical values reported represent mean values ± S.D. The statistical significance compared with control values was calculated using one-way ANOVA followed by Turkey’s post-test in the figures. * *p <* 0.05 and *** p* < 0.01 were considered statistically significant.

## 3. Results

### 3.1. Bovine Lactoferrin Inhibits Invasion of OSCC Cells

To determine the biological effects of bLF on the invasive properties of human OSCC cells, HSC2 cells, the intermediate state of epithelial-mesenchymal transition (EMT) with mesenchymal features, and HOC313 with EMT were used.

The migration assay and wound healing assay showed that bLF inhibited migration of OSCC cells in a dose dependent manner (Figure 1A; Appendix A). Furthermore, the invasion assay using the Boyden chamber method indicated that bLF significantly suppresses invasion of HSC2 and HOC313 cells with high invasion ability, in a dose-dependent manner (Figure 1B).

### 3.2. Bovine Lactoferrin Suppresses Invasion of OSCC Cells by Controlling AP-1 Activity to Suppress Expression of MMP-1 and MMP-3

AP-1 is classified as a transcription factor that contributes to basal gene expression [18] and is composed of heterodimers of Jun and Fos proteins through a basic region-leucine zipper (bZIP) [19]. AP-1-dependent activity is activated through two upstream signal transductions of JNK/c-Jun and the ERK/c-Fos signaling pathways [20]. Recently, AP-1 has been reported to control cell transformation, proliferation, and survival [21,22]. In addition, AP-1 was identified as a transcriptional factor of MMP-1 and MMP-3 [23].

To understand the mechanism by which bLF potentiates its effects on the production of MMP-1 and MMP-3 and their involvement in the invasion of OSCC cells through inhibition of AP-1, we investigated the effects of bLF on the ERK, JNK, c-Jun, and c-Fos pathways. The results indicated that bLF dose-dependently downregulated expression of phosphorylated ERK1/2 (p-ERK1/2), p-JNK, p-c-Jun, and p-c-Fos in both HSC2 and HOC313 cell lines (Figure 2A(a,b)). In addition, bLF reduced the expression of MMP-1 and MMP-3, both at mRNA and protein levels, in HSC2 and HOC313 cells (Figure 2B(a–c)).

Together, the results revealed that bLF suppressed transcriptional factor AP-1 activity through the inhibition of JNK/c-Jun and the ERK/c-Fos signaling pathways, and reduced the expression of MMP-1 and MMP-3 to control invasion of OSCC cells.

### 3.3. Bovine Lactoferrin Requires Low-Density Lipoprotein Receptor-Related Protein 1 (LRP1) to Inhibit OSCC Cell Invasion

LRP1, a receptor of bLF [24], participates in controlling cell invasion [25]. Thus, we assessed the expression of LRP1 in OSCC cells and examined its function in the inhibition of OSCC cell invasion by bLF. OSCC cells, HSC2, HOC313, and others, clearly expressed LRP1 (Figure 3A; Appendix A). Then, to examine the importance of LRP1 in the effects of bLF on OSCC cell invasion, LRP1 was depleted in HSC2 and HOC313 by siRNAs (Figure 3B). At the investigated time points of 12 h in HSC2 and 24 h in HOC313, bLF (100 μg/mL) interestingly failed to exert its additive effects on cell migration in LRP1 knockdown cells (Figure 3C; Appendix A). Similarly, bLF also did not show its protective effects on invasion of HOC313 in LRP1 depleted cells in the invasion assay (Figure 3D(a,b)). These results indicate that bLF inhibits invasion of OSCC through LRP1.

### 3.4. Bovine Lactoferrin Suppresses tPA-Induced Cell Invasion of OSCC by Reducing MMP-1 and MMP-3 Production

tPA, one of the precursors of the active serine protease plasmin, has been reported to increase invasion of cancer cells through the activation of some MMPs [26,27]. It is also known that tPA activates invasion of cancer cells via direct binding to LRP1 [28,29,30].

To clarify the importance of LRP1 in the induction and inhibition of invasion of OSCC, tPA was used as the antagonist of bLF. To identify the mechanism by which bLF-inhibited tPA induced invasion of OSCC cells, we activated OSCC cells using tPA (20 nM) and bLF (100 μg/mL). We found that tPA (20 nM) remarkably activated OSCC cells migration, about 25% (*p* = 0.009) at 6 h in HSC2 and 30% (*p* = 0.008) at 12 h in HOC313; however, bLF (100 μg/mL) strongly suppressed the tPA-induced migratory abilities of both examined cell lines (Figure 4A; Appendix A).

The same inhibitory effect trend of bLF on tPA-induced invasion of HSC2 and HOC313 cells was observed using the Boyden chamber method. We interestingly found that bLF (100 μg/mL) highly suppressed tPA-induced HSC2 and HOC313 cell invasion by about 90% (Figure 4B).

To further examine how bLF suppresses tPA-induced invasion of HSC2 and HOC313 cells, expressions of MMP-1 and MMP-3, as well as their signaling pathways, and transcriptional factor AP-1 were investigated. tPA drastically activated expression of p-ERK1/2/p-c-Fos and p-JNK/p-c-Jun (Figure 5A,B). bLF, however, inhibited these signaling pathway complexes, resulting in reduced AP-1 activity as a transcription factor of MMP-1 and MMP-3 both at mRNA and protein levels (Figure 5A,B). The reduction of tPA-induced AP-1 by bLF provoked the diminution of tPA-induced MMP-1 and MMP-3 expression at RNA and protein levels (Figure 5C(a–c)). These results suggest that bLF inhibits the invasion of OSCC cells by suppressing tPA-induced MMP expression.

## 4. Discussion

The interest in natural products has expanded in cancer therapy, and LF has been used in clinical trials [15,31]. Various in vitro and in vivo studies proved that bLF reduces cancer risks, but does not have any inhibitory effects on normal cells [10,11,15,24,32]. Moreover, bLF has been approved by the European Food Safety Authority, Drug Administration (USA), and the Therapeutic Goods Administration [33,34]. Its multi-functional properties gives bLF a remarkable impact against the invasion of cancers including OSCC.

LF, the second largest component of milk protein after casein, is used as a dietary supplement and drug [35]. Most orally administered LF is degraded by stomach fluid with only small amounts of intact LF reaching in the small intestine [24,36,37]. Therefore, maintaining bLF in intact form during absorption is crucial for it to exert its anti-cancer effects on treatment of OSCC. Liposomal bLF (LbLF), a multi-lamellar phospholipid vesicle containing a significant amount of bLF molecules, was reported to be an effective delivery system to prevent LF degradation in the stomach [38].

The MMPs family, including collagenases, stromelysins and stromelysin-like MMPs, matrilysins, gelatinases, MMP19-like MMPs, membrane-type MMPs (MTMMPs), and other MMPs, which is activated by various factors including growth factors, cytokines, physical stress, oncogenic transformation, cell-cell, and cell-ECM interactions, is capable of degrading essential components of the extracellular matrix (ECM), and is involved in tumor invasion and tumor metastasis [39]. Recently bLF was reported to inhibit the invasion of breast cancer cells via the inhibition of FAK phosphorylation at tyrosine (Tyr)-397 [40], and enhanced AMP-activated protein kinase activity (AMPK) thereby inhibited the expression of MMP-2 (Gelatinase A) and MMP-9 (Gelatinase B) [41].

MMP-1, MMP-3, and MMP-9 were found to contain AP-1 binding sites in the proximal and distal promoters; Jun and Fos interacted with the AP-1 cis-element and activated the transcription of the MMP genes [42]. On the other hand, promoter regions of the MMP-2 gene do not contain a conserved AP-1 element, but instead AP-2 [43]. In addition, a role for AP-1 in tissue expression has been demonstrated to enhance expression of the human plasminogen promoter to direct various cell-type specific activity [44].

Recently, our group found that bLF reversed the programing of epithelial-to- mesenchymal transition (EMT) to mesenchymal-to-epithelial transition (MET) in OSCC by decreasing the ERK signaling pathway, reducing the expression of TWIST, and improving membrane bound E-Cadherin [11]. Currently, evidence is accumulating that the ERK1/2/MMP-1 axis could inhibit metastasis of human oral squamous cells [45], and that knocking down of MMP-1 could suppress the PI3K/Akt/c-myc signaling pathway and EMT in colorectal cancer [46]. In addition, MMP-3 directly mediated EMT by upregulating TWIST, thereby disrupting E-cadherin expression [47,48]. Our previous and current findings may explain the inhibitory effects of bLF on invasion of OSCC by direct and indirect pathways. (i) bLF directly suppresses the transcriptional factor TWIST and improves a membrane bound E-cadherin, (ii) bLF may reduce MMP-1 or MMP-3 via an inhibition of AP-1 and disrupt E-Cadherin. To clearly address whether bLF indirectly reversed EMT to MET by the disruption of the AP-1/MMP-1/MMP 3 axis, and improved E-Cadherin expression in OSCC, further experiments are needed.

The process of invasion of OSCC requires a key mechanism of destruction of basement membranes, which is mediated by proteolytic enzymes including MMP-1 and MMP-3. Previously, substantial expression of MMP-1 and MMP-3 was observed in OSCC cases involving poor prognosis and metastasis, and this is associated with tumor progression and poor survival rates [49,50,51]. In the current study, we interestingly found that bLF reduced the invasion of OSCC cells via attenuation of MMP-1 and MMP-3 transcriptional levels through AP-1 (Figure 1A,B and Figure 2A–C).

The reduction of MMP-1 and MMP-3 expression was reported to correlate with the inflammatory cytokine IL-6, signal transducer and activator of transcription (STAT), and the AP-1 (IL-6/STAT-1/AP-1) axis in colon cancer. The IL-6 regulation of MMPs was identified by gamma-activated site (GAS)-like, STAT binding elements (SBEs) within the proximal promoters of the MMP-1 and MMP-3 genes, which is associated with the AP-1 components (c-Fos or Jun) [52]. Previously, our group found that inflammatory cytokines, including IL-1 β and IL-6, were highly expressed in cancerous cells [32]. Therefore, bLF suppression of IL-6 and the STAT-signaling pathway resulted in the attenuation of the invasion of OSCC [10,11], and may also be involved with the reduction of MMP-1 and MMP-3 expression.

LRP1, expressed by diverse cell types [53], is bound and endocytosed by over 30 structural ligands including lipoprotein, proteinase, proteinase-inhibitor, and plasminogen activators [54,55]. It is a 600 kDa protein, which is quickly cleaved by the enzyme furine convertase into two main domains, a 515 kDa extracellular-α-chain and an 85 kDa transmembrane intracytoplasmic β-chain [56]. These two domains are responsible for internalization and catabolism of various ligands including bLF. The internalization and catabolism of various biological components is involved in many physiological and pathological processes, including the invasive behaviors of various cancer cells [25,57,58]. It was previously confirmed that LRP1 is responsible for the invasion of cancer cells, but not in LRP1 silenced cells [28,29].

In our study, we observed that LRP1 was expressed in all examined cell lines (Figure 3A,B; Appendix A) indicating the involvement of LRP1 in the invasive behaviors of OSCC cells. The confirmation using siLRP1 clearly illustrated that LRP1 plays an important role in promoting the migration and invasion of OSCC cells (Figure 3C,D). These results are consistent with a previous report, which indicated that the high expression of LRP1 promoted breast cancer cell invasiveness [39,59]. However, it was also clarified that LRP1 may play a role as a suppressor receptor for the metastatic phenotype in melanoma in response to ApoE [60]. Regarding the controversy over the function of LRP1 in the invasiveness of cancer cells, we tested two antagonistic ligands, bLF and tPA, in order to gain insight into the detailed mechanism of LRP1 in OSCC cell invasion.

One ligand of LRP1, tPA, has been reported to trigger intracellular signal transduction via ERK1/2 through the induction of specific target genes’ production, including MMP-2 and MMP-9 [28,61]. Moreover, bLF was also reported to be a ligand of LRP1 by blocking LRP1-dependent stimulation of cholesteryl ester synthesis in cultured human fibroblasts and mediating endocytosis into the cells via LRP1 to inhibit endogenous signal transduction, including NF-κB and MAP kinase (MAPK) activation [24,62,63]. In this study, the findings that bLF mediated the inhibition of tPA-induced migration and the invasion of OSCC cells (Figure 4A,B; Appendix A; Figure 3A,B; Appendix A) indicate that bLF and LRP1 play an important role in the orchestration of LRP1, bLF, and tPA complexes. To confirm this notion, knockdown of LRP1 in OSCC cells showed that neither bLF nor tPA failed to exert their potential effects on invasion (Figure 3C,D; Appendix A). Furthermore, we determined how bLF reduces OSCC cell invasion by focusing on delineating the effects of the relationship between bLF and tPA on the expression of MMP-1 and MMP-3, and on their transcriptional factor AP-1. MMP-1 is highly expressed in OSCC and changes the behavior of cancer cells by promoting tumor cells’ invasion [64]. Moreover, the expression of MMP-1 is localized to the invasive edge of OSCC cells adjacent to stromal cells and is correlated with high invasiveness, lymphatic metastasis, and the poor prognosis of patients with OSCC [65,66]. In addition, MMP-3 is generated from plasminogen and expressed in the peri-tumoral area to facilitate tumor growth and invasion. Recently, it was found in high concentrations in the serum of OSCC patients and it plays an important role in inducing the migration and invasion of OSCC cells [65,67]. In the present study, we found that tPA strongly induced the migration and invasion of OSCC cells via suppression of AP-1 and upregulation of expression of both MMP-1 and MMP-3 (Figure 4C,D; Appendix A, indicating the amplification ability of tPA in proteolytic conversion of MMPs from their zymogen to active forms. Thus, this transformation may contribute to the malignant behaviors of OSCC. Interestingly, bLF was shown to reduce cell mobility and invasion, and inhibited tPA-induced AP-1, MMP-1, and MMP-3 expression (Figure 4A–C and Figure 5A–C).

For detailed mechanisms, we also illustrated how bLF attenuates AP-1 signaling pathways. We demonstrated earlier that AP-1 is involved in two signal transduction cascades, the JNK/c-Jun and the ERK/c-Fos. The intracellular signaling of these complexes is responsible for the downstream actions of tPA or bLF at their receptor LRP1. In this respect, we confirmed that tPA induced AP-1 activity at the investigated time point. In contrast, bLF drastically downregulated tPA-induced AP-1 (Figure 5A,B). Together, here we provide the first evidence that bLF suppresses tPA-induced OSCC cell invasion through MMP-1 and MMP-3 induction via the deactivation of AP-1 complexes.

## 5. Conclusions

In summary, to the best of our knowledge, this is the first report showing the effect of bLF on migration and invasion of OSCC cells through the reduction of tPA-induced AP-1, MMP-1, and MMP-3 (Figure 6). This study could provide a forward direction in the field of LF research and bLF could be considered as a potential molecule in OSCC prevention and/or therapy.

## Figures and Tables

**Figure 1 pharmaceutics-15-00562-f001:**
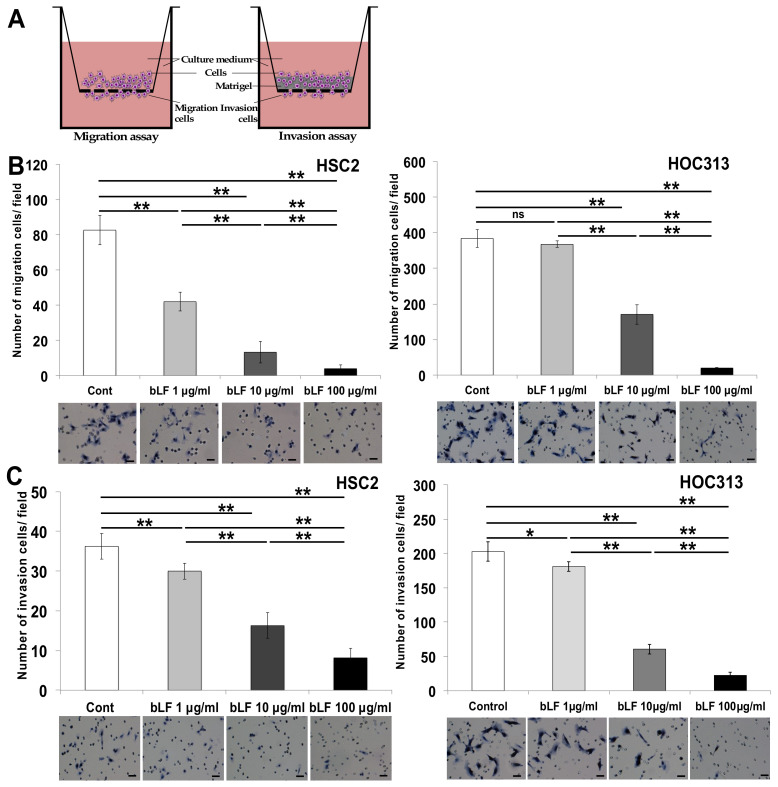
Bovine lactoferrin suppressed invasion of OSCC cells. For the migration assay, 6 × 10^5^ cells were plated into 6 cm dishes. The medium was changed after 24 h of incubation and pretreated with bLF (1 μg/mL, 10 μg/mL, 100 μg/mL) for 72 h before trypsinizing. (**A**) Schematic representation of chamber migration and invasion assays. (**B**) The effects of bLF on HSC2 cells were investigated at 6 h; suppression of HOC313 cell migration at 12 h was also evaluated. (**C**) Invasion assay using the Boyden chamber with its specific analogue. Seventy-two hour bLF-pretreated HSC2 and HOC313 cells (2 × 10^4^) were cultured in a culture insert with an 8 μm pore size in the presence or absence of bLF (1 μg/mL, 10 μg/mL, 100 μg/mL). Invasive cells were stained by haematoxylin and photographed under a phase contrast microscope. Data (n = 3) were triplicated and are represented as mean ± S.D. Statistical analysis was performed using one-way ANOVA. * *p* < 0.05 and ** *p* < 0.001 compared with control (0 μg/mL of bLF) group. Scale bar 100 μm.

**Figure 2 pharmaceutics-15-00562-f002:**
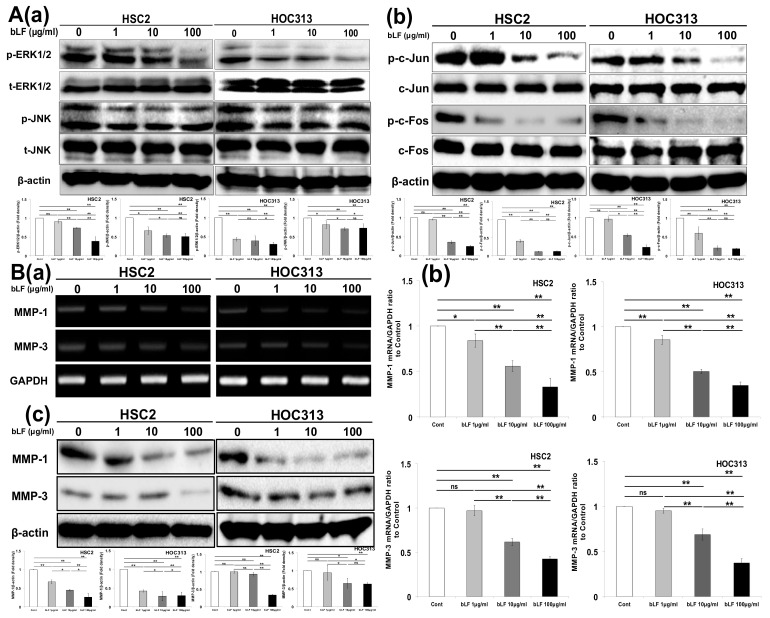
Bovine lactoferrin reduced MMP-1 and MMP-3 expression through attenuation of AP-1 activity. HSC2 and HOC313 cell lines were cultured into 6-well plates for mRNA analysis and 6 cm plates for protein analysis. (**A**(**a**)) Protein expression of ERK1/2 and JNK cells treated with and without bLF for 24 h, (**b**) Expression of c-Jun and c-Fos in cells after bLF application for 24 h. (**B**) Inhibition of MMP-1 and MMP-3 mRNA production using (**a**) RT-PCR and (**b**) qRT-PCR after bLF treatment for 24 h (right panels), and (**c**) MMP-1 and MMP-3 protein expression after bLF application for 48 h (left panel). The experiments were conducted at least 3 times. ** p* < 0.05; **** *p* < 0.01.

**Figure 3 pharmaceutics-15-00562-f003:**
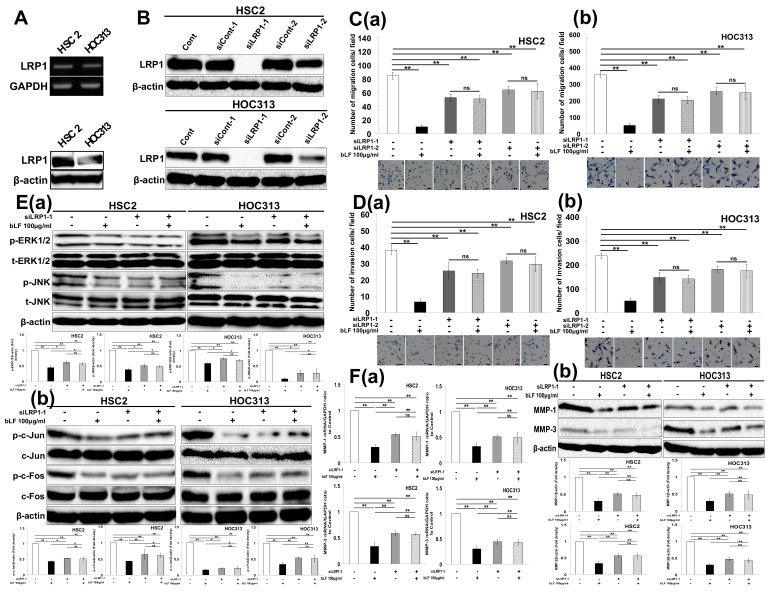
Bovine lactoferrin inhibited invasion of OSCC cells through LRP1. HSC2 and HOC313 cells were seeded into 6-well plates and 6 cm dishes for mRNA and protein extraction, respectively. (**A**) mRNA and protein expressions of LRP1 in OSCC cell lines. (**B**) Confirmation of knockdown capacity of LRP1 in observed OSCC cells. (**C**) Investigation of the function of bLF and LRP1 on the migration, (**a**) HSC2 cells and (**b**) HOC313 cells (**D**) invasion of OSCC cells, (**a**) HSC2 cells and (**b**) HOC313 cells. (**E**) Observation of the role of bLF and function of LRP1 in (**a**) ERK1/2, JNK, and (**b**) AP-1 signaling pathways. (**F**) Investigation on effects of bLF and involvement of LRP1 in MMP-1 and MMP-3 expression by (**a**) qRT-PCR and (**b**) western blot. The data are presented as the mean ± S.D. ** p* < 0.05; **** *p* < 0.01. Typical data of 3 independent experiments (n = 3) were investigated. ns: not significant. Scale bar 100 μm.

**Figure 4 pharmaceutics-15-00562-f004:**
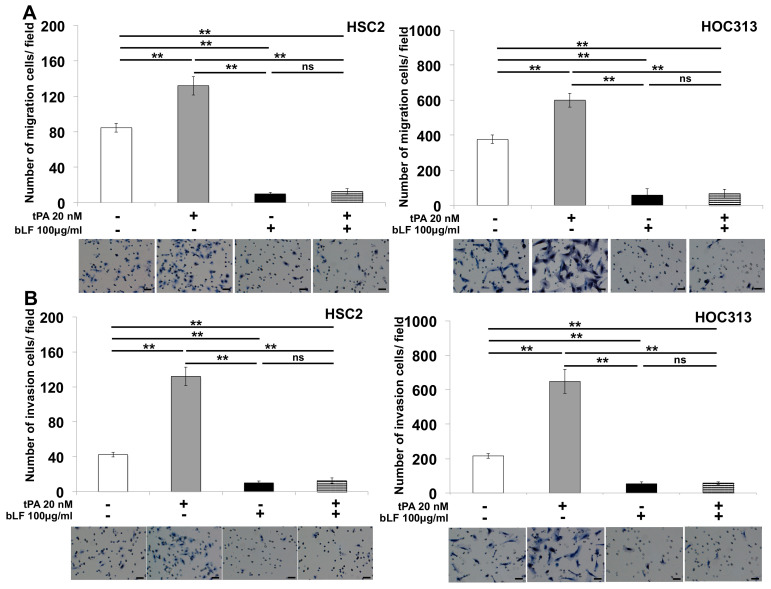
Bovine lactoferrin inhibited tPA−induced OSCC cell invasion. tPA−induced migration of OSCC cells was evaluated using migration assays. tPA (20 nM) and bLF (100 μg/mL) were used as described in the Materials and Methods. (**A**) HSC2 and HOC313 cells (6 × 10^5^) were cultured with or without tPA and/or bLF and trypsinized cells were subjected to perform the migration assays. The migrated cells were examined 6 h (for HSC2) and 12 h (for HOC313) after stimulation. (**B**) The invasion of HOC313 cells was investigated using the Matrigel invasion assay, in which cells stimulated with and without tPA (20 nM) and/or bLF (100 μg/mL) were allowed to migrate through a Matrigel-coated Transwell insert for 24 h. n = 3 in triplicate; the mean ± S.D. ** *p* < 0.01. ns: not significant. Scale bar 100 μm.

**Figure 5 pharmaceutics-15-00562-f005:**
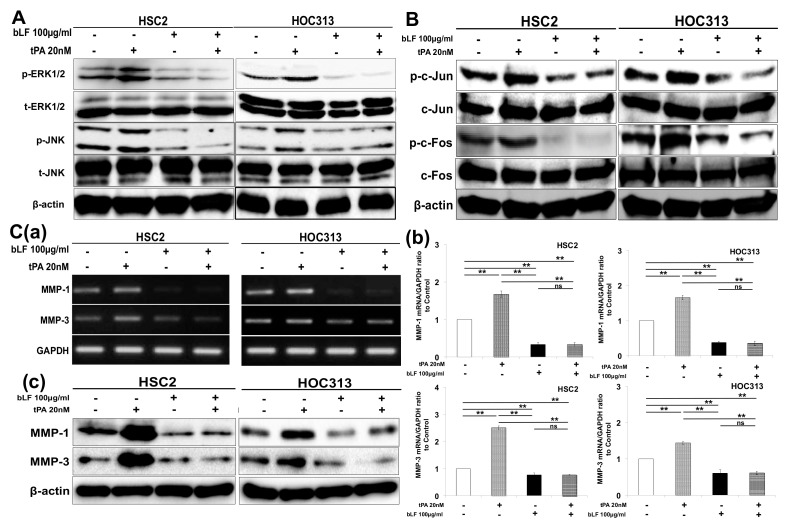
Bovine lactoferrin downregulated tPA−activated AP-1 to suppress expression of MMP−1 and MMP−3. HSC2 and HOC313 cells were plated into 6−well plates and 6 cm dishes for mRNA and protein analysis, respectively. (**A**,**B**) bLF (100 μg/mL) and tPA (20 nM) were applied to cultured plates containing HSC2 and HOC313 cells and incubated for 24 h. Proteins were extracted, and the effect of bLF on tPA−induced p-ERK1/2, p−JNK, p−c-Jun, and p-c-Fos was analyzed by western blotting. (**C**) Cells were stimulated using tPA (20 nM) and bLF (100 μg/mL) for 24 h for mRNA and 48 h for protein expression. Cells were then harvested and the expression levels of MMP−1 and MMP−3 were analyzed by (**a**) RT-PCR (left panel), (**b**) qRT−PCR (right panels), and (**c**) western blotting (left panel). All experiments were performed in triplicate (n = 3). ns: not significant. ** *p* < 0.01. ns: not significant.

**Figure 6 pharmaceutics-15-00562-f006:**
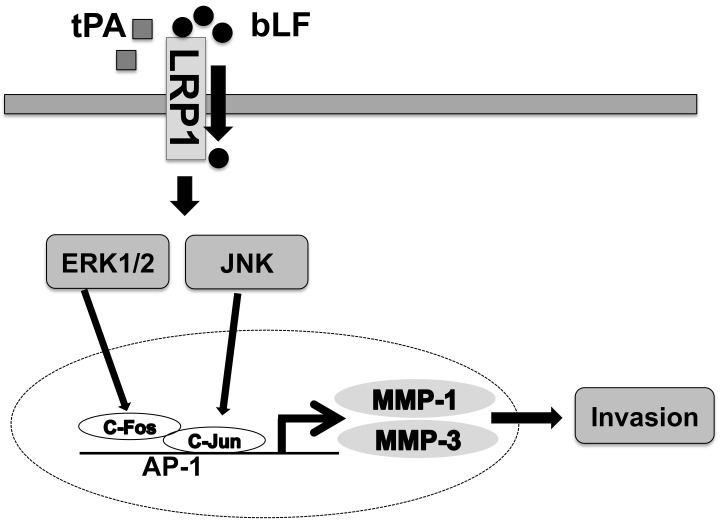
Schematic representation of the inhibitory effect of bLF on invasion of OSCC and its underlying molecular mechanisms. bLF and tPA are ligands for LRP1. The competitive binding of bLF, and internalization of bLF through LRP1, suppresses ERK1/2 and JNK and tPA-induced ERK1/2 and JNK signaling pathways. The inactivation of ERK and JNK is responsible for the attenuation of AP-1 (complex of c-Fos and c-Jun), thereby inhibiting MMP-1 and MMP-3 induction. Suppression of MMP-1 and MMP-3 indicates the suppression of OSCC cell invasion by bLF.

## Data Availability

Not applicable.

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
