# Peer review of "Molecular Mechanisms of Inhibitory Effects of Bovine Lactoferrin on Invasion of Oral Squamous Cell Carcinoma"

_pharmaceutics, 2023, doi:10.3390/pharmaceutics15020562_

Round 1

Reviewer 1 Report

The manuscript provides an interesting insight on the molecular mechanism of Lactoferrin for oral SCC therapy.

I would suggest some clarification/ additions to further improve the manuscript:

1.      Is there any data on 3D SCC model

2.      What dose is recommended to achieve the required therapeutic effect (inhibition) , is it given systemically or topically?

3.      If LF is to be used locally or topically for inhibiting SCC , what formulation is appropriate for that? Is it to be used as such or in liposomal system or conjugated form?

4.      Please provide a schematic diagram describing the chamber migration and invasion assay.

5.      Was there a difference between the free Lf and liposomal LF regarding the cellular efficacy.

6.      The manuscript focus on the pharmacology of LF on SCC cells, the authors should describe at least a delivery system or recommend  future direction of this research from pharmaceutics / drug delivery point of view to fit the scope of the journal.

7.      The first paragraph in the discussion is not appropriate to be included as such and should be deleted.

Author Response

pharmaceutics-2133461

Dear referees;

Thank you very much for your consideration on our manuscript to be a major revision in pharmaceutics, and we would like to show our gratitude for your valuable comments. Based on your suggestions we revised our mistakes, modify the figures, added rational points to the study, and include most of data into our manuscript. After revision, we added some data as following:

A list of the major changes made

  • We enlarged Figures 1-5 and rearranged spaces in each figure
  • We added schematic diagram of migration and invasion assays in Figure 1A.
  • We provided more information into discussion section.
  • The drug safety of bovine lactoferrin (bLF) to normal oral cells and its approvals.
  • A recommended delivery system of bLF for a future direction of bLF application.
  • The purposes of The MMP1 and MMP3 were examined in this manuscript.
  • More possible mechanisms of EMT involving in invasion of OSCC for further directions of LF to cancer-research field.

Dear Reviewer # 1

Remark: The manuscript provides an interesting insight on the molecular mechanism of Lactoferrin for oral SCC therapy. I would suggest some clarification/ additions to further improve the manuscript:

Reviewer’s comments:

1- Is there any data on 3D SCC model?

Response 1: Thank you very much for your indication.

            We are so sorry that we did not perform any experiment using 3D SCC model using HSC2 and HOC313 cell lines in this manuscript. Instead, previously, our group investigated the effect of bLF on invasions of OSCC using both in vitro and in vivo models (Chea C et al., 2018).

2- What dose is recommended to achieve the required therapeutic effect (inhibition), is it given systemically or topically?

Response 2: We appreciate your indication.

            To the best of our knowledge local administration of bLF is easier to imagine in the clinical setting, since in this study bLF directly affects cancer cells as an anti-neoplastic agent. However, we recommend systemically application of bLF. Bovine LF (bLF), a protein found in cow’s milk, did not show toxic effects and is safe in doses lesser than 2000 mg/kg body weight/day in adults, according to the European Food Safety Authority (EFSA) Panel. We previously confirmed that orally applied liposomal bLF (L-bLF) successfully suppressed periodontal bone destruction in periodontitis rat model and pathological progression of arthritis in Rheumatoid arthritis mouse model (Yamano E et al., 2010; Inubushi T et al., 2012; Yanagisawa S et al., 2022). Moreover, actually bLF has been clinically used in various cancer patients like colon cancer with successful inhibitory effects and in other clinical trial (Tsuda H et al., 2010; Ishikado A, et al., 2010). We believe that supplement, which patients can take freely, is most suitable application for OSCC patients.

3- If LF is to be used locally or topically for inhibiting SCC, what formulation is appropriate for that? Is it to be used as such or in liposomal system or conjugated form?

Response 3: Thank you very much for your questions.

            Local administration of bLF is much simpler to apply in term of SCC therapy. However, the safety of local administration of bLF in vivo has not been confirmed, so the possibility of local administration needs to be examined in the future, including the devising of administration methods. On the other hand, there have been numerous studies on systemic administration of bLF, and no major side effects have been reported. We have also confirmed the safety and efficacy of liposomal bLF, which was designed to avoid degradation of bLF in the stomach and promote absorption of intact bLF in the intestine, in studies of periodontal disease and rheumatoid arthritis. Therefore, we are considering systemic administration of liposomal bLF first. In the future, we believe that it is necessary to devise a way for systemically administered bLF to accumulate in cancer tissues.

4- Please provide a schematic diagram describing the chamber migration and invasion assay.

Response 4: We appreciate your comment.

            We prepare the schematic diagram of both assays as shown in Fig. R1. We add this schematic diagram in Figure 1A.

Fig. R1

5- Was there a difference between the free Lf and liposomal LF regarding the cellular efficacy.

Response 5: We are very sorry for our mistakes of indicating Liposomal bLF (LbLF) in the manuscript. In fact, we did not use liposome to wrap around bLF in vitro study; we used LbLF in vivo study (Chea C et al., 2018). The purpose of using the L-bLF is to include large amounts of bLF in multilayers of liposomes to avoid degradation by stomach enzymes and to increase the amount of intact bLF entering the intestinal tract (Ishikado A et al., 2005).

6- The manuscript focus on the pharmacology of LF on SCC cells, the authors should describe at least a delivery system or recommend future direction of this research from pharmaceutics / drug delivery point of view to fit the scope of the journal.

Response 6: We would deeply thank you for pointing out the important point. We add potential delivery system of bLF in the discussion of the manuscript.

7- The first paragraph in the discussion is not appropriate to be included as such and should be deleted.

Response 7: Thank you for your indication. We remove this part out from the manuscript.

Reviewer 2 Report

General comments:

This study reported the importance of LRP1 in bLF-suppressed invasion of OSCC cells 27 via reduction of AP-1 and MMP production. However, there are some concerns need to be attentioned.

Major comments:

1. The drug safety of bovine lactoferrin to normal oral cells were not examined or discussed.

2. The MMP1 and MMP3 were examined by mRNA and protein. Why the authors did not examined the MMP2 and MMP9 by zymography to detect the migration activity?

3. EMT signaling was not addressed in this migration study.

Minor comments:

1. Most of the figures are too small to see. Please enlarge them because there are space in the left side.

2. Some ? marks appear in Figure 3. Please correct them.

Author Response

pharmaceutics-2133461

Dear referees;

Thank you very much for your consideration on our manuscript to be a major revision in pharmaceutics, and we would like to show our gratitude for your valuable comments. Based on your suggestions we revised our mistakes, modify the figures, added rational points to the study, and include most of data into our manuscript. After revision, we added some data as following:

A list of the major changes made

  • We enlarged Figures 1-5 and rearranged spaces in each figure
  • We added schematic diagram of migration and invasion assays in Figure 1A.
  • We provided more information into discussion section.
  • The drug safety of bovine lactoferrin (bLF) to normal oral cells and its approvals.
  • A recommended delivery system of bLF for a future direction of bLF application.
  • The purposes of The MMP1 and MMP3 were examined in this manuscript.
  • More possible mechanisms of EMT involving in invasion of OSCC for further directions of LF to cancer-research field.

Dear Reviewer # 2

General comments:

This study reported the importance of LRP1 in bLF-suppressed invasion of OSCC cells via reduction of AP-1 and MMP production. However, there are some concerns need to be attentioned.

Major comments:

1- The drug safety of bovine lactoferrin to normal oral cells were not examined or discussed.

Answer 1: Thank you for your important suggestion.

We add more information about drug safety of bovine lactoferrin and its effects on normal oral cells into the manuscript.

2- The MMP1 and MMP3 were examined by mRNA and protein. Why the authors did not examine the MMP2 and MMP9 by zymography to detect the migration activity?

Answer 2: We do appreciate for your crucial indication. We will add rational explanations in our manuscript.

            We purposely selected human MMP-1 and MMP-3 in our studies, (i)-MMP-1, MMP-3, and MMP-9 were found to contain AP-1 binding site in the proximal and specifically in distal promoters; Jun and Fos interact to the AP-1 cis-element and activate the transcription of the MMP gene (Gutman A et al., 1990). On the other hand, promoter regions of MMP-2 gene do not contain a conserved AP-1 element (Frisch SM et al., 1990). (ii) bLF was reported to suppress MMP-2 and MMP-9 expression in breast cancer (Fayard B et al., 2009; Rodriguez-Ochoa N et al., 2022) (iii) MMP-1 and MMP-3 correlate with poor prognosis in esophageal associated with lymph node metastasis (Murray GI et al., 1998; Shima I et al., 1992).

            We did not investigate migration activity using zymography in this manuscript. Zymography is more specific to measure proteolytic activity of gelatinases including gelatinase A (MMP-2) and gelatinase B (MMP-9). However, we focused on MMP-1 in the study, which have collagenolytic activity; MMP-3, a stromelysin subgroup and potent activator of MMP-1, hydrolyses components of fibrolytic system including fibrinogen, plasminogen activator (Moilanen M et al., 2003; Lijnen HR et al., 2000).    

3- EMT signaling was not addressed in this migration study.

 Answer 3: We would deeply thank you for pointing out the important points. We address more EMT signaling in the manuscript.

Minor comments:

1- Most of the figures are too small to see. Please enlarge them because there are space in the left side.  

Answer 1: Thank you very much for your fruitful suggestion. We enlarge all figures for a better quality of the article.

2- Some ? marks appear in Figure 3. Please correct them.

 Answer 2: We are so sorry for the mistakes. We correct them and add more description into Figure 3 legend.

Reviewer 3 Report

In this manuscript, the authors aimed to investigate the effects of Lactoferrin on invasion of OSCC and its underlying molecular mechanism. Their results indicated that bLF can suppress invasion of OSCC cells down-regulate the expression of MMP-1 and MMP-3 via LRP1 and AP-1. Overall, the study was well designed and the results were clearly presented, but there are still some points should be addressed before it can be accepted for publication.

1. The authors used 1, 10, and 100ug/ml bLF to treat OSCC cells. Why did they select these 3 concentrations? According to their published paper, 100ug/ml bLF can significantly inhibit cell proliferation and promote cell apoptosis. Was the inhibition of invasion and migration of bLF on OSCC cells cause by its pro-apoptotic effect?

2. The authors indicated that they used “HSC2 cells without epithelial-mesenchymal transition (EMT) and HOC313 with EMT”. What is the difference between these two cell lines? But the effect of bLF on these 2 cell lines were almost same.

3. The rational of this study was not clearly stated. Why did not select the AP-1? And why MMP-1 and MMP-3 were selected but not other MMPs?  

4. In Figure 2B, the bands were not clear and qRT-PCR was strongly suggested for better quantification.

Author Response

pharmaceutics-2133461

Dear referees;

Thank you very much for your consideration on our manuscript to be a major revision in pharmaceutics, and we would like to show our gratitude for your valuable comments. Based on your suggestions we revised our mistakes, modify the figures, added rational points to the study, and include most of data into our manuscript. After revision, we added some data as following:

A list of the major changes made

  • We enlarged Figures 1-5 and rearranged spaces in each figure
  • We added schematic diagram of migration and invasion assays in Figure 1A.
  • We provided more information into discussion section.
  • The drug safety of bovine lactoferrin (bLF) to normal oral cells and its approvals.
  • A recommended delivery system of bLF for a future direction of bLF application.
  • The purposes of The MMP1 and MMP3 were examined in this manuscript.
  • More possible mechanisms of EMT involving in invasion of OSCC for further directions of LF to cancer-research field.

Dear Reviewer # 3

In this manuscript, the authors aimed to investigate the effects of Lactoferrin on invasion of OSCC and its underlying molecular mechanism. Their results indicated that bLF can suppress invasion of OSCC cells down-regulate the expression of MMP-1 and MMP-3 via LRP1 and AP-1. Overall, the study was well designed and the results were clearly presented, but there are still some points should be addressed before it can be accepted for publication.

1- The authors used 1, 10, and 100μg/ml bLF to treat OSCC cells. Why did they select these 3 concentrations? According to their published paper, 100μg/ml bLF can significantly inhibit cell proliferation and promote cell apoptosis. Was the inhibition of invasion and migration of bLF on OSCC cells cause by its pro-apoptotic effect?

Response 1: We would deeply thank you for pointing out the critical points.

  • Bovine lactoferrin (bLF) 1, 10 and 100 μg/ml was selected according to previous reports and appropriate to optimal concentration of bLF used in clinical trials (180mg/day) (Tsuda H et al., 2010; Ishikado A 2010 ). bLF 1 and 10μg/ml are the minimal doses; and bLF 100μg/ml is a higher dose. bLF 14.5 mg/ml (175 μM) was also used as the highest dose in in-vitro study (Pereira CS et al., 2016); Costantino E et al., 2022), however, we did not used this extreme high concentration in this manuscript.
  • bLF 100 μg/ml was reported to inhibit proliferation and induce apoptosis in OSCC cells (Chea C et al., 2018). The inhibition on growth of OSCC cells was observed at day 2 (48h) in gene level and phenotype level at days 4-6 (96-144 h). To observe the inhibition of bLF on growth of OSCC cells, which is independent from migration and invasion of OSCC cells, we observed the effects of bLF on proliferation of HSC2 and HOC313 cells in 6, 12, 24, and 48 h. The results showed that phenotypically, no significant reduction in number of HSC2 and HOC313 cells were observed R1. In contrast, in this manuscript, we observed phenotypically the inhibitory effects of bLF on migration and invasion of OSCC cells in 6 h for HSC2 and 12 h for HOC313 cells.  

Fig. R1

2- The authors indicated that they used “HSC2 cells without epithelial-mesenchymal transition (EMT) and HOC313 with EMT”. What is the difference between these two cell lines? But the effect of bLF on these 2 cell lines were almost same.

Response 2: Thank you for your suggestion.

            HSC2 cell line, is a Head and Neck Squamous Cell Carcinoma (HNSCC), has an intermediate state of Epithelial-Mesenchymal-Transition (EMT) or a partial EMT (p-EMT) and acts like cancer cells with mesenchymal behavior; it does not completely lose their epithelial features (Pastushenko I, et al., 2018). Unlike HOC313 cell line has EMT features including loss E-cadherin expression and increase metastasis (Nguyen PT et al, 2013). Cells with p-EMT pose a higher metastatic risk rather than complete EMT ones (Shao W et al., 2021). We add more details about the differences between the behaviors of these two investigated cell-line in the manuscript.

3- The rational of this study was not clearly stated. Why did not select the AP-1? And why MMP-1 and MMP-3 were selected but not other MMPs?

Response 3: Thank you for your comments. We will add the rational explanations and selection of MMP-1 and MMP-3 in this study but not other MMPs in the discussion.

4- In Figure 2B, the bands were not clear and qRT-PCR was strongly suggested for better quantification.

Response 4: We do appreciate for your crucial indication. In Figure 2B, we elucidate the expressions of MMP-1 and MMP-3 using (a) RT-PCR, (b) qRT-PCR, and (c) western blotting. We modify some descriptions in figure legend of Figure2B for a better understanding.

Round 2

Reviewer 2 Report

All reviewer's concerns were well responded.

Reviewer 3 Report

All my concerns were well addressed and the revision was improved. I think it is fine for publication now.